# Impact of the COVID-19 pandemic on neonatal admissions in a tertiary children's hospital in southwest China: An interrupted time-series study

Weiqin Liu[1,2], Qifen Yang[3], Zhen-e Xu[1,2], Ya Hu[1,2], Yongming Wang[1,2], Zhenqiu Liu[1,2], Qianqian Zhao[1,2], Zhuangcheng Wang[4], Hong Wei[1,2], Ziyu Hua[1,2]*

1 Department of Neonatology, Children's Hospital of Chongqing Medical University, National Clinical Research Center for Child Health and Disorders, Ministry of Education Key Laboratory of Child Development and Disorders, Chongqing, China, 2 Chongqing Key Laboratory of Pediatrics, Children's Hospital of Chongqing Medical University, Chongqing, China, 3 School of Life Sciences, Southwest University, Chongqing, China, 4 Children's Medical Big Data Intelligent Application Chongqing University Engineering Research Center, Chongqing, China

* h_ziyu@126.com

**Data Availability Statement:** All relevant data are within the manuscript and its Supporting Information files.

## Abstract

### Background

The unprecedented coronavirus disease 2019 (COVID-19) pandemic has caused millions of infections worldwide and represents a significant challenge facing modern health care systems. This study was conducted to investigate the impact of lockdown measures in a tertiary Children's Hospital in southwest China, which might be used to predict long-term effects related to health-seeking behavior of parents/caregivers.

### Methods

This study included newborns enrolled over a span of 86 weeks between January 4, 2019, and August 27, 2020. We designated two time periods for analysis purposes: a stable pre-COVID period(55 weeks between January 4, 2019, and January 23, 2020) and a COVID-impacted period (31 weeks between January 24, 2020, and August 27, 2020). An interrupted time-series analysis was employed to compare changes and trends in hospital admissions and disease spectra before and after the period of nonpharmaceutical interventions (NPIs). Furthermore, this study was conducted to evaluate whether the health-seeking behavior of parents/caregivers was influenced by pandemic factors.

### Results

Overall, 16,640 infants were admitted to the neonatology department during the pre-COVID period (n = 12,082) and the COVID-impacted period (n = 4,558). The per week neonatal admissions consistently decreased following the first days of NPIs (January 24, 2020). The average weekly admission rates of 220/week pre-COVID period and 147/week COVID-impacted period. There was an evident decrease in the volume of admissions for all disease

**Funding:** this study was supported by grants from the Clinical Research Project of Children's Hospital of Chongqing Medical University (YBXM 2019-007), the funders had no role in study design, data collection and analysis, decision to publish, or preparation of the manuscript.

**Competing interests:** NO authors have competing interests.

spectra after the intervention, whereas the decrease of patients complaining about pathological jaundice-related conditions was statistically significant ($p<0.05$). In the COVID-impacted period, the percentage of patients who suffered from respiratory system diseases, neonatal encephalopathy, and infectious diseases decreased, while the percentage of pathological jaundice-related conditions and gastrointestinal system diseases increased. The neonatal mortality rates (NMRs) increased by 8.7% during the COVID-impacted period compared with the pre-COVID period.

## Conclusions

In summary, there was a significant decline in neonatal admissions in a tertiary care hospital during the COVID-19 Pandemic and the associated NPIs. Additionally, this situation had a remarkable impact on disease spectra and health-seeking behavior of parents/caregivers. We, therefore, advise continuing follow-ups and monitoring the main health indicators in vulnerable populations affected by this Pandemic over time.

## Introduction

In December 2019, an outbreak of pneumonia caused by coronavirus disease 2019 (COVID-19) was first reported in Wuhan, China [1]. COVID-19, caused by severe acute respiratory syndrome coronavirus 2 (SARS-CoV-2), was declared a pandemic by the World Health Organization on March 11, 2020 [2]. As of this writing (December 1, 2020), COVID-19 infection has been confirmed in nearly 63 million individuals worldwide and caused at least 1,400,000 deaths [3].On January 23, 2020, the Chinese government raised its national public health response to the highest state of emergency in Wuhan, China, to control the rapid spread of this viral disease. The government has enacted a series of lockdown strategies, including city-wide lockdown, screening measures at train stations and airports, school closures, and suspended nonessential businesses [4].

Nonpharmaceutical interventions (NPIs) cause a novel challenge for healthcare systems. In recently published studies, NPIs effects have caused a noticeable reduction in both adult and pediatric patient volumes during the COVID-19 Pandemic [5–7]. While newborns are at low risk for severe disease and death from COVID-19, any impacts on their health outcomes will likely be attributable to the indirect effects of the Pandemic on health systems [8, 9]. However, to the best of our knowledge, no reports have described changes in the behavior of the parents/caregivers. This study was conducted to investigate the impact of NPIs on neonatal admissions and disease spectra, which might be used to predict long-term effects related to health-seeking behavior of parents/caregivers.

## Methods

### Study design

This was a single-center, retrospective, observational study that used collected patient data from the Department of Neonatology, Children's Hospital of Chongqing Medical University, Chongqing, China. The patients' medical records were accessed from January 4, 2019, to August 27, 2020, and data were anonymized at time of access. Ethics approval was obtained from the Children's Hospital of Chongqing Medical University Human Research Ethics Committee (Approval No. 2020206–1). Regarding informed consent, the ethics committee waived

the requirement for informed consent; however, confidentiality was maintained. The study followed the Strengthening the Reporting of Observational studies in Epidemiology guidelines (STROBE) [10].

## Setting

The Department of Neonatology of the Children's Hospital of Chongqing Medical University is a tertiary care hospital and the most extensive local neonate referral center in southwestern China. This facility offers a comprehensive range of services to the districts of Chongqing and neighboring provinces, including Guizhou, Sichuan, Yunnan, and Tibet, and serves approximately 350,000 babies in the associated area. The Department of Neonatology has 310 beds and provides services that account for nearly 10,000 hospital admissions each year over the last 5 years. Babies can be admitted to this neonatal unit from 3 sources: Inter-hospital transportation (outborn), Emergence(outborn), and Outpatients (outborn).

Dedicated transport teams are staffed separately from NICU personnel specifically for the purpose of transport of patients to and from the hospital. These personnel do not have patient assignments. Our neonatal unit receives direct referrals from the labour ward or Level I or II neonatal care services using the transport system. "Emergency" means that critically ill patients were transferred and admitted into our neonatal unit directly via outpatients.

## Participants

This study included newborns enrolled over a span of 86 weeks between January 4, 2019, and August 27, 2020. Demographic and clinical data were retrospectively collected from the hospital electronic record system (EPR). There were no exclusion criteria.

## Study variables

The variables selected to describe the demographic and clinical information of the cases were the volume of admissions and neonate transport, sex, age (days), admission, length of hospital stay (days), treatment, neonatal mortality rates (NMRs), gestational age (weeks + days), birth weight (gram) and disease spectra, weekly admission rates, admissions visiting from out-of-province.

As a unique group characterized by immaturity, neonates are susceptible to multiorgan disorders (e.g., complicated pneumonia and jaundice, complicated sepsis, and necrotizing enterocolitis). At discharge, medical staff assigned a primary diagnosis based on all available clinical data. The classification of our disease spectra was based on the primary diagnosis at discharge in the medical record. The gestational age and birth weight of missing values were deleted (n = 49 and n = 8, respectively).

## Intervention

The Chongqing government initiated the first-level response to major public health emergencies and the implementation of the NPIs on January 24, 2020 [11]. After the start of the NPIs, the hospital established a special isolation ward that was equipped with 8–16 beds, 5–8 trained nurses, and 2 pediatricians. The ward was equipped with a ventilator including invasive and noninvasive (CPAP and NIPPV) ventilation. During the Pandemic, there was no requirement to reduce neonatal transfers, except for out-of-province transfers.

We designated two periods for analysis purposes: a stable pre-COVID period, reflecting the routine workload of the Department of Neonatology (55 weeks between January 4, 2019, and

January 23, 2020), and a COVID-impacted period (31 weeks between January 24, 2020, and August 27, 2020). A one-week time unit was chosen to provide optimal precision to the model.

### Outcomes

The primary outcome was to detect changes in the levels and trends of hospital admissions and disease spectra before and after the period of NPIs and assess the impact of the control measures. Second, this study was conducted to evaluate whether the health-seeking behavior of parents/caregivers was influenced by pandemic factors.

### Statistical analysis

Interrupted time series (ITS) is a suitable model for assessing interventions' short- and long-term effects in a quasi-experimental study. The segmented regression model is one of the most commonly used methods for ITS methods [12]. The following segmented linear regression model was applied.

$$Yt = \beta 0 + \beta 1^* \text{ time} + \beta 2^* \text{ intervention} + \beta 3^* \text{ post-slope} + \varepsilon t$$

The outcome variable (Yt) was the weekly number of cause-specific hospital admissions. Time (in weeks) was treated as a continuous variable indicating the number of weeks from the observation time to time t. Intervention was a dummy variable indicating the pre-COVID period (coded 0) or the COVID-impacted period (coded 1). Post-slope is a continuous variable, where pre-COVID period is code 0 and COVID-impacted period takes the same value as the time variable. $\beta 1$ represents the estimate at the baseline level of the outcome at time zero before the NPIs; $\beta 2$ estimates the level of change in the rate immediately after the NPIs; $\beta 3$ represents changes in trends after the NPIs, and $\varepsilon t$ is the error term of the model [13, 14]. The Durbin–Watson (DW) method was used to detect and exclude the time series data stability. Values approaching 2 and 4 represent almost no autocorrection, whereas 0 indicates positive autocorrection. The generalized linear model (Prais-Winsten estimates) was used to determine the most efficient way to estimate the values.

Descriptive analyses, including the median and interquartile range as appropriate, were performed to describe the characteristics of infants during the pre-COVID and COVID-impacted period. Wilcoxon sum rank tests, Mann–Whitney tests, and Pearson chi-squared tests were used where applicable to compare demographic and outcome characteristics between the pre-COVID and COVID-impacted period. A *p*-value less than 0.05 was considered statistically significant.

All analyses were fit in RStudio 1.0.44 (RStudio, Inc.) using R v.4.0.2 [15].

## Results

The admissions per week of neonates were analyzed between January 4, 2019, and August 27, 2020. Overall, 16,640 infants were admitted to the neonatology department during the pre-COVID period (n = 12,082) and the COVID-impacted period (n = 4,558). The per week neonate admissions consistently decreased following the first days of the NPIs (January 24, 2020). The average weekly admission rates of 220/week pre-COVID period and 147/week COVID-impacted period (S1 File). With the mitigation of the Pandemic, the results showed a slow recovery of increase in the admissions per week of the neonatal department(*p*<0.05). Nevertheless, more time is required to return to the pre-COVID period levels of admissions (Fig 1).

No significant differences in gestational age or birth weight were observed between groups. However, the admission pathway changed. The percentage of emergencies and inter-hospital transfers increased, while the outpatient ratios decreased. The patients seeking consultations in the COVID-impacted period were younger than those seeking consultations in the pre-

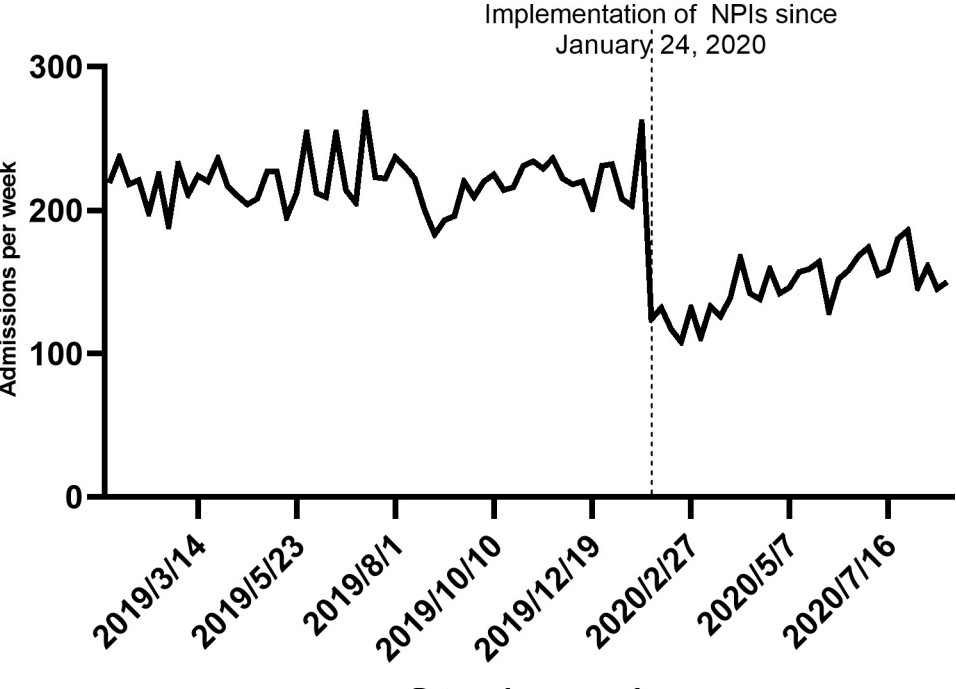

**Fig 1. Weekly volume of neonatal admissions visits between January 4, 2019, and August 27, 2020.**

COVID period (median age 5 days *vs*. 6 days, $p<0.05$). This is likely because those admitted via outpatients were reduced. In the COVID-impacted period, 58.8% of all visits were by children aged ≤7 days old, compared with 54% during the pre-COVID period. In the pre-COVID period, patients were hospitalized for a median of 6 days (*IQR*: 4–10 days), which increased to 7 days (*IQR*: 5–13 days) in the COVID-impacted period ($p<0.05$). The comparison of the two periods in terms of patients' demographic characteristics is shown in Table 1 and S1 File.

After 1-order differencing was used, the different sequence tended to be stationary. All of the Durbin-Watson statistics approached 2, indicating that there was no autocorrelation in the observations. When the intervention was implemented on January 24, 2020, the volume of neonatal transports and admissions declined sharply ($p<0.05$). With the improvement of the pandemic situation, the observations showed a slow-growth trend for the volume of neonatal transports and admissions ($p<0.05$) (Table 2 and Fig 2).

There was an evident decrease in the volume of admissions/week for all disease spectra after the intervention. Additionally, there was a statistically significant decrease in the number of patients with pathological jaundice-related conditions ($p<0.05$) (Table 2 and Fig 2).

However, their recovery trend was faster than that other disorders ($p<0.05$), therefore relatively speaking the proportion of admissions attributable to jaundice increased.

The treatment/week volume is significantly decreased, except therapeutic hypothermia. The average weekly therapeutic hypothermia rates of 1.0 /week pre-COVID period and 1.4/week COVID-impacted period. The percentage of therapeutic hypothermia cases increased, while mechanical ventilator ratios decreased. The percentage of surgical and exchange transfusions was unaffected by these measures. NMRs increased by 8.7% during the COVID-impacted period compared with the pre-COVID period. However, no statistical differences were found between NMRs ($p = 0.66$) (Table 1).

**Table 1. Demographic characteristics for the patients.**

| Variable | | Pre-COVID | COVID impacted | p value |
|---|---|---|---|---|
| Total,n | | 12082 | 4558 | |
| Average weekly admission rates,n | | 220 | 147 | |
| Gestational Age (Weeks+days), n (%) | ≤27+6 | 201 (1.65) | 74 (1.63) | 0.34 |
| | 28+0–31+6 | 715 (5.93) | 249 (5.48) | |
| | 32+0–36+6 | 2775(23.00) | 1056 (23.30) | |
| | 37+0–41+6 | 8322 (69.10) | 3154(69.44) | |
| | ≥42+0 | 38 (0.32) | 7 (0.15) | |
| Birth weight(gram) median (IQR) | | 3060 (2530–3410) | 3070 (2550–3440) | 0.66 |
| Age (days), median (IQR) | | 6 (1–17) | 5 (1–14) | <0.05* |
| Number of patients, n (%) | ≤7 (days) | 6613 (54.7) | 2682 (58.8) | <0.05* |
| | >7 (days) | 5469 (45.3) | 1876 (41.2) | |
| Gender, n (%) | Male | 6772 (56.1) | 2674 (58.7) | <0.05* |
| | Female | 5310 (43.9) | 1884 (41.3) | |
| Admissions,n (%) | Emergency | 1147 (9.5) | 611 (13.4) | <0.05* |
| | Outpatient | 6695 (55.4) | 2060 (45.2) | |
| | Inter-hospital transport | 4240 (35.1) | 1887 (41.4) | |
| Admissions visiting from out-of-province, n (%) | | 2225 (18.4) | 702 (15.4) | <0.05* |
| Average weekly Disease spectra rates, n (%) | Respiratory system | 76.2 (34.7) | 47.5 (32.3) | <0.05* |
| | Infectious diseases | 12.5 (5.7) | 6.6 (4.50) | |
| | Gastrointestinal system | 27.5 (12.5) | 19.5 (13.3) | |
| | Pathological jaundice-related diseases | 58.7 (26.7) | 45.6 (31.0) | |
| | Neonatal encephalopathy | 4.1 (1.87) | 2.5 (1.73) | |
| | Others | 40.7 (18.53) | 25.3 (17.17) | |
| Average rate of weekly treatment, n (%) | Surgery | 9.1 (4.12) | 6.9 (4.70) | <0.05* |
| | Exchange transfusion[a] | 1.2 (2.05) | 1.0 (2.19) | |
| | Mechanical ventilator[b] | 19.9 (26.1) | 9.9 (20.9) | |
| | Therapeutic hypothermia[c] | 1.0 (23.1) | 1.4 (55.7) | |
| Length of hospital stay (days), median (IQR) | | 6 (4–10) | 7 (5–13) | <0.05* |
| Average rate of weekly Fatality, n (‰) | | 2.3 (10.3) | 1.6 (11.2) | 0.66 |

Median (interquartile range, IQR) for continuous variables, number (%) for categorical variables.

a Percentage of Exchange transfusion in the Pathological Jaundice-Related Diseases groups.

b Percentage of Mechanical ventilator in the Respiratory system groups.

c Percentage of Therapeutic hypothermia in the Neonatal encephalopathy groups.

Statistical differences:

* $p < 0.05$.

## Discussion

Chongqing, one of the municipalities in China, is among the high-risk regions for SARS-CoV-2 spread in China. As of this writing (December 1, 2020), Chongqing has reported 590 confirmed cases and 6 COVID-19-associated deaths. As the Chongqing government implemented a series of NPIs after January 24, 2020, these measures resulted in a significant reduction in hospital admissions [11]. This is consistent with published data from other cohorts (not including newborn data) in many countries that show a range of declines in hospital admissions. In the United States, visits and access to the Department of Veterans Affairs hospitals declined by 42% in the COVID-19 period [16]. In France, the number of pediatric emergency department (PED) visits and related hospital admissions after the NPIs decreased by 68% and

**Table 2. Estimated coefficients of the segmented regression model for neonate admissions and disease spectra since January 4, 2019–August 27, 2020.**

| Variable | Coefficient | Estimate | Std. Error | *t*-value | *P*-value |
|---|---|---|---|---|---|
| **Neonatal admissions** | Intercept β0 | 217.226 | 4.631 | 46.903 | <0.05 * |
| | Baseline trend β1 | 0.089 | 0.144 | 0.618 | 0.54 |
| | Level change after intervention β2 | −176.566 | 24.723 | −7.142 | <0.05* |
| | Trend change after intervention β3 | 1.408 | 0.369 | 3.816 | <0.05* |
| **Neonatal Transport** | Intercept β0 | 53.837 | 2.293 | 23.476 | <0.05* |
| | Baseline trend β1 | 0.018 | 0.071 | 0.254 | 0.80 |
| | Level change after intervention β2 | −52.845 | 12.295 | −4.298 | <0.05* |
| | Trend change after intervention β3 | 0.553 | 0.183 | 3.026 | <0.05* |
| **Respiratory system** | Intercept β0 | 72.994 | 4.067 | 17.944 | <0.05* |
| | Baseline trend β1 | 0.130 | 0.125 | 1.042 | 0.30 |
| | Level change after intervention β2 | −37.848 | 21.337 | −1.774 | 0.08 |
| | Trend change after intervention β3 | 0.031 | 0.321 | 0.099 | 0.92 |
| **Pathological jaundice-related diseases** | Intercept β0 | 56.345 | 2.374 | 23.735 | <0.05* |
| | Baseline trend β1 | 0.081 | 0.073 | 1.105 | 0.27 |
| | Level change after intervention β2 | −57.833 | 12.633 | −4.578 | <0.05* |
| | Trend change after intervention β3 | 0.582 | 0.188 | 3.084 | <0.05* |
| **Gastrointestinal system** | Intercept β0 | 26.837 | 1.620 | 16.561 | <0.05* |
| | Baseline trend β1 | 0.023 | 0.050 | 0.476 | 0.63 |
| | Level change after intervention β2 | −16.461 | 8.653 | −1.903 | 0.06 |
| | Trend change after intervention β3 | 0.104 | 0.129 | 0.807 | 0.42 |
| **Infectious diseases** | Intercept β0 | 10.484 | 1.173 | 8.938 | <0.05* |
| | Baseline trend β1 | 0.071 | 0.036 | 1.963 | 0.05 |
| | Level change after intervention β2 | −9.867 | 6.158 | −1.602 | 0.11 |
| | Trend change after intervention β3 | 0.012 | 0.092 | 0.133 | 0.90 |

Statistical differences:

* *p*< 0.05.

45%, respectively [7]. With the abatement of the pandemic, the neonatal admission volume slowly recovered (*p*<0.05). Nevertheless, more time is required for admission volumes to return to the pre-pandemic levels.

The reason for the observed reduction in neonate visits after the strictest NPIs were implemented remains unclear. There is a common understanding that hospital admissions decreased due to the most stringent NPIs, as individuals were worried about becoming infected by SARS-CoV-2 in the hospital [17]. Patients with "mild cases" were more prepared to seek treatment at a primary care facility rather than at a tertiary care hospital, provided that medical resources were available. On the other hand, due to the development of the Internet, parents can address some or even all of their newborns' problems through effective online counseling, resulting in a smaller volume of neonatal outpatient visits.

This report showed that the number of admissions for related respiratory system diseases (e.g., neonatal pneumonia, acute respiratory distress syndrome) and infectious diseases (e.g., neonatal sepsis, meningitis) decreased after public health interventions. This may be partially attributed to the reduction in social activities and the emphasis on the use of personal protective equipment (e.g., hand-washing, wearing a mask) [18]. However, the proportion of gastrointestinal diseases and surgeries were unaffected by these measures. This is partly because gastrointestinal disorders, especially anomalies, are relatively more common in our tertiary care hospital (e.g., megacolon, esophageal atresia, pyloric hypertrophic obstruction). Some of

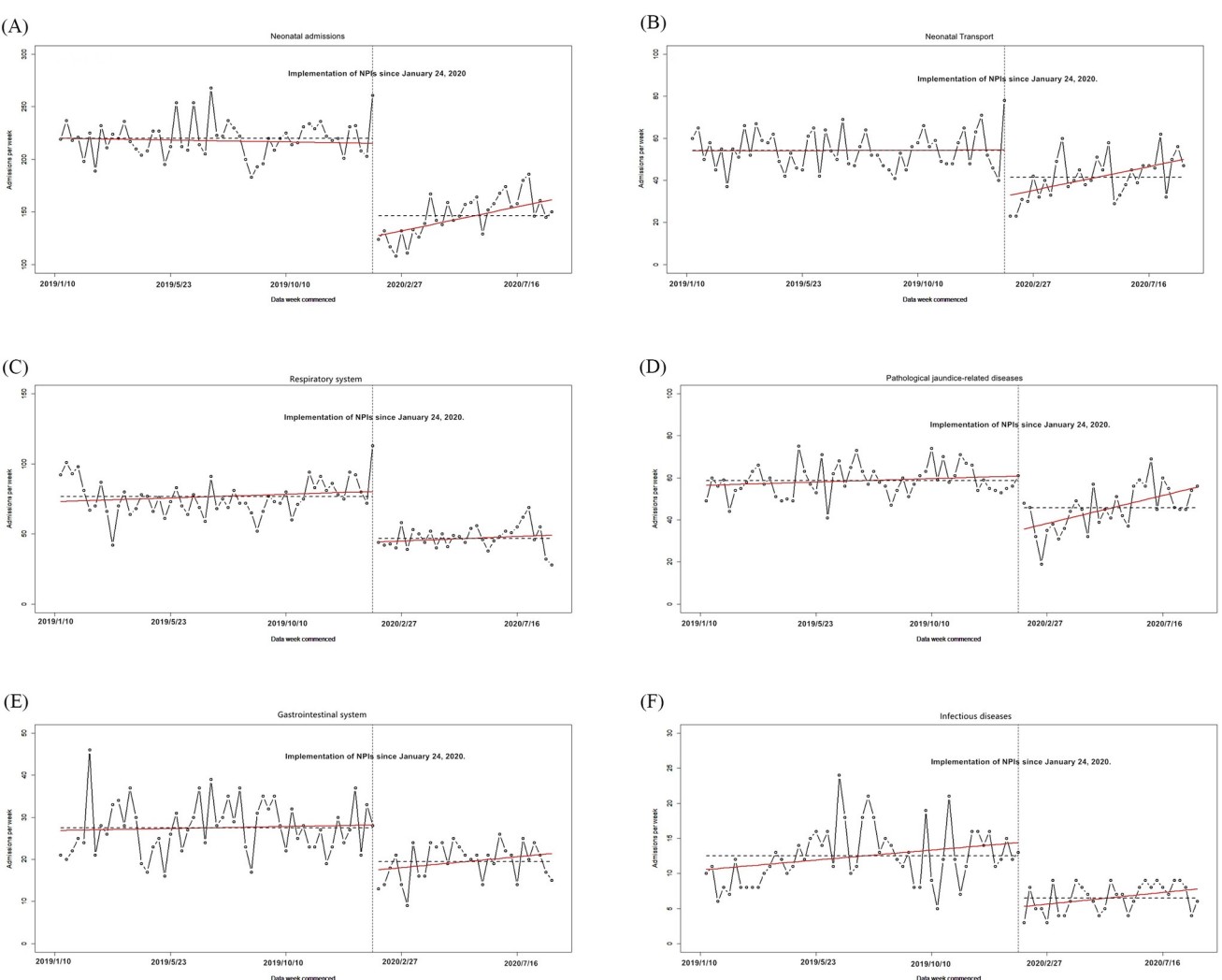

**Fig 2. Impact of interventions on attendances and disease spectra based on segmented regression analysis.** (A) Neonatal admissions (B) Neonatal Transport (C) Respiratory system (D) Pathological jaundice-related disease (E) Gastrointestinal system (F) Infectious diseases.The black points in the plot represent weekly changes in the volume of attendances and disease spectra. The red fitted lines are from Prais-Winsten estimates. Dashed horizontal lines indicate the level change.

these diseases require surgical intervention in the neonatal period. As one of the regional neonate emergency care providers and the most extensive local neonate referral center, only the children's hospital could perform a sequence of operations.

Neonatal hyperbilirubinemia is a common condition, affecting more than 60% of full-term and 80% of preterm infants [19–21]. It was hypothesized that NPIs might not have a noticeable effect on the volume of pathological jaundice-related disease cases admitted to neonatal centers. However, our single-center study showed a significant decrease in jaundiced patients' admissions when public health interventions were implemented. This phenomenon might be partly attributed to early and stringent preventive measures and partly due to parents' insufficient awareness of pathological hyperbilirubinemia (jaundice may be a "mild" disease, which primary care facilities can also treat). However, the rebounding trend in the number of jaundiced patients' admissions ($p<0.05$) was associated with the relaxation of public health interventions and families seeking better medical resources.

Although there were no significant differences between gestational age and birth weight, the results showed an elevated proportion of therapeutic hypothermia during NPIs. One probable reason for such unacceptable poor quality of maternity care and poor neonatal health outcome is changes in accessing emergency obstetric and /or neonatal cares which include delays in reaching health facilities (delays at the journey) and delays in receiving appropriate care (delays at the health facilities) [22]. Another possible reason is maternal factors. Some pregnant women reduced the frequency of antenatal visits due to the fear of contracting COVID-19, and some pregnant women were never examined from the initiation of NPIs until the delivery of their baby. Therefore, some potential risk factors may not have been detected in a timely manner, leading to fetal distress and necessitating emergency delivery [17].

The data suggested that NMRs might be increased in COVID-impacted period. Preterm birth is the leading cause of neonatal mortality and morbidity in both high- and low-income countries [23]. No significant differences were observed between the two groups regarding premature birth. Therefore, this increase in mortality rate appears to be driven by other factors and is not likely to be explained by preterm birth. It is more likely that changes in illness severity, health-seeking behavior of parents/caregivers, and inadequate medical resources factors result in this phenomenon.

The admissions for neonatal encephalopathy(NE) from 4/wk to 2.5/wk–of those admitted with NE–23% required therapeutic hypothermia compared to 56% COVID-impacted period–suggesting those babes were sicker on admission. The neonatal period is classified as early (the first 7 days) and late (the remaining 21 days) neonatal period, and majority neonatal mortality occur in the early neonatal period [24, 25]. The proportion of children aged ≤7 days old was higher in the COVID-impacted period(54.7% pre-COVID and 58.8% COVID-impacted period). In China, the implementation of NPIs also results in severe restrictions affecting the health-seeking behavior of parents/caregivers. There was concern that parents/ caregivers stay at home to care for her baby with progressive disease rather than come to the hospital where the risk of contracting COVID-19 was perceived to be higher. The Department of Neonatology has insufficient critical care staff during the COVID-impacted period (70 residents/week pre-COVID period and 48 residents/week COVID-impacted period). It is possible that reduced capacity of healthcare services to provide timely diagnoses and treatment. Reassuringly, these mortality analyses found no statistical differences ($p = 0.66$). These results should be interpreted with caution, and additional studies are needed to clarify whether our observations represent a significantly higher mortality rate adjusted for disease severity or whether they merely reflect increased primary care management of less severe conditions.

## Limitations

There are several potential limitations of this study that should be addressed. First, a main drawback is that this study was derived from a single tertiary care center with a retrospective design and a relatively small sample size. Therefore, it does not have strong universal validity, especially for primary health care. Second, this study was observational, and the association between the decreased volume of neonate admissions and NPIs was not necessarily causal. The birth rate in Chongqing has been declining since 2018. The Office for National Statistics (ONS) output reported a birth rate of 11.02‰ in 2018 and 10.48‰ in 2019. Perhaps this declining trend of the volume of neonate admissions is merely a manifestation of the declining birth rate or has been aggravated by the implementation of NPIs. This topic needs further exploration of other databases to overcome the above limitations. Third, interrupted time series regression analyses are regarded as the strongest "quasi-experimental" approach and are particularly useful when an RCT is infeasible. This finding has to be interpreted with caution

because the study design did not investigate seasonal effects. Use of many treatments varies seasonally because of cyclic variations in the illnesses for which they are prescribed. Accounting for seasonally correlated errors usually requires at least 24 monthly data points. Interrupted time series regression analyses models we discussed assume a linear trend in the outcome within each segment. The assumption of linearity often may hold only over short intervals. Changes may follow non-linear patterns [12]. Fourthly, we did not follow-up on the long-term effects of the "missing" visits and the adverse effects that are associated with them. Finally, The COVID-19 Pandemic has drastically impacted on entire health economy. For instance, there are potential barriers to accessing obstetric and neonatal health care services, barriers to good healthcare-seeking behavior of parents/ caregivers, transportation barriers, and parents/ caregivers' financial barriers(housing instability, food insecurity). Further research should be conducted to elaborate on this aspect.

## Conclusions

In summary, there was a significant decline in neonatal hospital admissions in a tertiary care hospital during the COVID-19 Pandemic and associated NPIs. Additionally, this situation had a remarkable impact on disease spectra. Our findings also raise concerns about the healthcare-seeking behavior of parents/ caregivers. It is possible that an inappropriate choice of not referring a baby to the hospital who needs medical attention has a long-term negative impact on neonatal health. We, therefore, advise continuing follow-ups and monitoring the main health indicators in vulnerable populations living in areas affected by this Pandemic over time.

## Supporting information

**S1 File. Demographic characteristics for the patients.**
(ZIP)

## Acknowledgments

We acknowledge all the colleagues, both physicians, and nurses, that their hard work made possible the exceptional hospital activity capable of taking care of hundreds of patients a day. Without the work of all those people, this article could not be written.

## Author Contributions

**Conceptualization:** Weiqin Liu, Qifen Yang, Ziyu Hua.

**Data curation:** Zhen-e Xu, Ya Hu, Yongming Wang, Zhenqiu Liu, Qianqian Zhao.

**Formal analysis:** Zhuangcheng Wang.

**Funding acquisition:** Ziyu Hua.

**Methodology:** Zhuangcheng Wang.

**Resources:** Zhenqiu Liu, Qianqian Zhao.

**Software:** Zhuangcheng Wang.

**Supervision:** Hong Wei.

**Visualization:** Weiqin Liu.

**Writing – original draft:** Weiqin Liu, Qifen Yang, Ziyu Hua.

**Writing – review & editing:** Ziyu Hua.

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
