## [Decision Letter · Decision Letter 0]

15 Mar 2021

PONE-D-21-02629

The impact of COVID-19 pandemic on neonatal admission: an interrupted time-series study

PLOS ONE

Dear Dr. Ziyu Hua,

Thank you for submitting your manuscript to PLOS ONE. After careful consideration, we feel that it has merit but does not fully meet PLOS ONE’s publication criteria as it currently stands. Therefore, we invite you to submit a revised version of the manuscript that addresses the points raised during the review process.

I am returning your manuscript with three reviews. Please pay attention to the following reviewer suggestions and give them due consideration. Please take the following points particularly into account: 

A thorough language revision of the manuscript is needed to ensure clarity and consistency. Please also double-check any numbers and calculations presentedPlease indicate the form of consent obtained (written/oral) or the reason that consent was not obtainedConsider including measures such as neonatal mortality, gestational age, birth weight and diagnosis of neonatal encephalopathy in the analysis; if not please state why notMethods should be reported in accordance with the Strengthening the Reporting of Observational Studies in Epidemiology (STROBE) statementA discussion on the negative impact on infant outcomes is warrantedPlease elaborate more on the limitations of the chosen analytical method e.g. seasonal considerations

Please submit your revised manuscript within fourty five days od the date of this decision. If you will need more time than this to complete your revisions, please reply to this message or contact the journal office at plosone@plos.org. Please include the following items when submitting your revised manuscript:

We look forward to receiving your revised manuscript.

Kind regards,

Maria Moitinho de Almeida

Academic Editor

PLOS ONE

Journal Requirements:

2. In your ethics statement in the manuscript and in the online submission form, please ensure that you have discussed whether all data/samples were fully anonymized before you accessed them and/or whether the IRB or ethics committee waived the requirement for informed consent. If patients' guardians provided informed written consent to have data/samples from their medical records used in research, please include this information.

3. In the ethics statement in the manuscript and in the online submission form, please provide additional information about the patient records/samples used in your retrospective study, including the date range (month and year) during which patients' medical records/samples were accessed.

Reviewers' comments:

Reviewer's Responses to Questions

**Comments to the Author**

1. Is the manuscript technically sound, and do the data support the conclusions?

Reviewer #1: Partly

Reviewer #2: Yes

Reviewer #3: Yes

2. Has the statistical analysis been performed appropriately and rigorously? 

Reviewer #1: Yes

Reviewer #2: Yes

Reviewer #3: Yes

3. Have the authors made all data underlying the findings in their manuscript fully available?

Reviewer #1: No

Reviewer #2: Yes

Reviewer #3: No

4. Is the manuscript presented in an intelligible fashion and written in standard English?

Reviewer #1: Yes

Reviewer #2: Yes

Reviewer #3: Yes

5. Review Comments to the Author

Reviewer #1: This is a study reporting trends in admission rates, source of admission and diagnostic categories for babies admitted to a large tertiary newborn care unit in Southern China. Data have been derived from routine hospital data systems. Authors have used a time series analysis to compare two time periods: (i) pre-covid pandemic non pharmaceutical intervention implementation and (ii) covid pandemic non pharmaceutical interventions. Trends in hospital use, particularly those relating to newborns as a result of the pandemic have not frequently been reported. Clear drop in admission rates are demonstrated following the implementation of lockdown and other associated measures. Reductions were also seen in proportion of babies transferred in from neighbouring hospitals.

Major comments

These represent interesting descriptive data, however would be much richer if measures such as neonatal mortality, gestational age, birth weight and diagnosis of neonatal encephalopathy were included. It would be unusual for these measures, in particular mortality, not to be available from routine hospital data and therefore it seems odd not to have reported these. For instance the authors comment in the discussion that medical complexity and severity increased in those admitted in the second time period – but provide no data with which to show this. Tantalisingly they also comment that rates of therapeutic cooling increase but do not comment on diagnostic rates of neonatal encephalopathy (a sentinel marker of quality of perinatal maternal care). Please clearly state available measures from the HIS and why you have chosen the ones you have chosen.

Authors need to rewrite sections of the methods to describe more clearly the setting. They should not assume understanding or prior knowledge of this region, especially as lockdown and infection rates have been vastly different across the globe.

I think this paper could be improved enormously with inclusion of broader measures of newborn diagnosis and outcomes.

Methods need to set the scene much more clearly – for example, what is the population size covered by this unit (which from the description of newborn interventions provided would not be described as low resource). What is the usual care fatality rate for this unit? How premature are the babies who are cared for? What is the newborn birth rate within the hospital? It is not clear at all what admissions from outpatients refers to – please describe this more clearly. Anecdotally from my own work in Hong Kong a number of babies with prolonged jaundice and poor weight gain were admitted via outpatients to the NICU – not typical of UK settings. Is this similar in your hospital?

Methods shound be reported in accordance with the Strengthening the Reporting of Observational Studies in Epidemiology (STROBE) statement.

Also authors need to set the scene in the methods describing what policies were put in place by the local hospital systems due to COVID – for example, were hospitals told to reduce transfers? What were the pathways to admit from a referring hospital? Do you have isolation rooms etc? what lockdown/ NPIs were put in place and how did these change over time after the 24th? What are the staffing levels in the unit? How many ventilated beds/cots do you have? Did that change during the lockdown period?

Results comment around proportion of emergency and inter-hospital transfers increasing (lines 132) is confusing. Would be easier to understand if the methods included a brief description of where babies can be admitted from, eg. “babies can be admitted to this neonatal unit from 3 sources: labour ward/maternity care (inborn), outborn (transfers in from other hospitals) and outpatients (outborn).”

Univariate analysis appear to have been unnecessarily repeated – eg table 1 – chi2 should be conducted for one group of categorical outcomes rather than for each individual one. Please only present a single p value therefore for the group.

Would have been useful to present % increase or decrease by week of admission rates for example.

Results – major concern is exclusion of case fatality rate data and diagnosis of neonatal encephalopathy.

More detail can be given to jaundice- were these increases in pathological early jaundice, physiological jaundice or prolonged jaundice?

Discussion needs editing to explore data in more detail. For example what could the impact of this been on the wider health economy? did more babies die? What is the projected change in admissions like over time? When do you expect to reach previous levels? Or will you? Has the healthcare delivery in other settings/ home improved? Were some of the previous admissions inappropriate/unnecessary? More thought needs to be given to potential generalisability eg lines 201-202 – this is simply not true of the typical newborn population and likely related to the fact you are a tertiary surgical centre.

Discussion needs considerable rewriting to revisit some of the statements that are not evidenced through the data presented here.

Line 219 – I would argue that this is the most interesting data from this paper and least explored. As discussed above – what were the rates of neonatal encephalopathy diagnosis/ of deaths? Can you comment on reduced ventilated babies – is this because there were fewer premature babies (something reported in other countries) – can you test that within your data by looking at gestational age on admission?

Lines 223-224 – what is your evidence to back this statement up?> how have medical resources been more efficiently allocated? Might you want to continue this efficient allocation in the future (assuming you can evidence this statement). How has this allocation of resources impacted upon neonatal mortality and morbidity and on parental experience of healthcare? And on maternal outcomes? Were you able to measure and report on stillbirth rates for example?

Lines 224-225 – you don’t present any data to back this statement up in this paper. I have no doubt that has happened but it appears outkeeping with the data.

Lines 227-230 – you do not present any evidence to back this statement up at all.

Lines 231-232 – this is likely true, but how do your data help with this? Admission rates alone are not sufficient measure (and in fact one could argue an extremely poor measure) of efficacy of the healthcare system.

Lines 232-233 – this is also undoubtedly true – but how does this relate to your data?

Lines 238-239 – arguably this is a moderately well resourced unit – more detail needs ot be provided to allow the reader to compare resource of this unit to other settings.

Lines 241-243 – not really, you don’t show change in newborn behaviour.

Need to explore the what next? What additional data would be useful here with which to explore some of these hypotheses? For example from the wider health economy/ qualitative data from parents/ healthcare workers?

Conclusion – poorly written

General comments

Newborns arguably are unable to behave in any particular way – rather the healthcare systems and their parents/carers/ maternity services vary. Please rephrase “newborn healthcare behaviour” (used throughout the paper).

Ensure term pandemic rather than epidemic is used throughout

Minor comments

• Long and short title need to state location

Abstract

• background –needs to state where this study has been conducted

• methods – needs to include the time periods and which lockdown interventions and when

• methods - Define what the pre-COVID and COVID-impacted periods are. Seems pre-COVID period is very short

• conclusion – insufficient – needs to comment on fact that admissions reduced dramatically, these data only capture a small part of the picture – what happened to the non-admitted babies – did overall infant mortality rate increase for example?

Introduction

More information on background literature needs to be included here. For example a modelling study of this threat predicted between 250000-1.1 million extra neonatal deaths as a result of decreased service provision and access in LMICs. https://www.medrxiv.org/content/10.1101/2021.01.06.21249322v1

Methods:

Do HIS statistics capture all neonatal admissions for the region –and which region, how big, what population covered.

Statistical methods do not match

Introduction

Line 51 – “in” not “since”

Line 55 – “caused” not “causes”

Line 60 – consider abbreviation for NPI – non pharmaceutical interventions and then use this raterh than lockdown throughout

Line 63 “describing .. changes in newborn care” rather than “on the..”

Line 66 – see comment re phrase newborn medical behaviour being in appropriate for this age group

Methods

Line 71 – do you mean 10,000 babies per year over the last 5 years? Or 10,000 over the last 5 years? Unclear, please edit for sense.

Line 74 – no need for “besdies”

Line 91 – see comment re phrase newborn medical behaviour being in appropriate for this age group

Line 112 – I would not reloy on differentiating btween p values 0.05 or 0.01.

Methods- please describe how and when the diagnoses were given to the babies and by whom.

Results

Line 123 – replace visits with admissions

Lines 121-127 – please use phrase admissions per week (if this is what you are referring to) and edit this section for sense. For example – phrase “then it was reduced to a nadir of 108 people…” is not clear – do you mean it reduced to an admission rate of 108 babies/ week or that there were only 108 babies resident on the unit for that particular week?

figure 1 – please include dates on x axis, and also edit y axis to more accurately represent this measure (which I think is admissions per week)

figure 1 – annotate the figure to include what the dotted line indicates rather than just including this in the footnotes

line 132 – pandemic, not epidemic

line 132 – do not jump to causality here.

Lines 155-156 do not make sense

Lines 156-157 – when did a relatively high propotion of babies present with jaundice related conditions?

Figure 2 – please clarify and annotate as per figure 1

Discussion

Lines 177-178 edit for sense. Also highlight that these other cohorts do not include newborn data

Line 182 – pandemic not epidemic

Line 182 – edit to read admissions rather than entries

Line 195 – please include this detail in the methods

Line 206 – be more specific here between physiological/ pathological jaundice

Line 217 – how might you have measured degree of severity?

Line 218 – pandemic not epidemic

Reviewer #2: The authors conducted a rigorous ITS analysis of changes in NICU encounters pre- and post-COVID19 at a major children's hospital. The changes in encounter volume are generally in line with other observed decreases across the world. The stratified analyses are probably useful (though this reviewer is not an expert in NICU volume/demand).

While the study is statistically impeccable and the writing is very clear, more discussion of the negative impact on infant outcomes seems warranted. Similarly, if there has not been a commensurate increase in negative outcomes does that suggest many NICU admissions are unnecessary? The decrease in volume by itself is not that interesting. One approach might be a parallel ITS of negative outcomes with the same interruptions specified.

Reviewer #3: The topic of this article is interesting and relevant to the public most especially at this period. However, there are few issues that could use some improvement.

In the abstract the authors stated that “Overall, 13,540 infants were admitted to the NICU during the pre-COVID period (n= 12082) and COVID-impacted period (n=4558).”

The figures provided in line 34 – 35 do not add up and the statement is not clear on the disparity between the 13,540 and the figures provided in parentheses. It is also not clear what NICU means at this stage.

In line 64 it seems that the impact of lockdown measures ( such as curfew, opening and closing hours of business, who and where one could travel to ) on neonatal admissions and disease spectrum are to be studied. However, that seems not to be the case. The analysis and results presented seem to have studied if there is changes in neonatal admission and diseases spectrum pre-COIVD and during the COVID period. Or are the authors referring to the period of lockdown as the “lockdown measures”? This should be clarified in the manuscript.

In line 77 it was stated the “clinical and demographics information were collected…” while in line 78 though attempt was made to list the clinical information but the lists provided in line 79 does not differentiate between clinical and demographic information.

The authors should consider formatting the date on line 84 and 85 to commonly used time format for example (yyyy/m/d – yyyy/m/d).

In the manuscript in line 87 to 91 the outcomes and dependent variables were not clearly stated. Clarity is needed for the readers.

The use of interrupted time-series and the choice of break point seem appropriate. However, the authors should consider a change point analysis to detect a change in the data generating process. This might as well provide the basis for choosing or confirming the choice of the break point used in this analysis. In addition, time series analyses may be confounded by seasonal effects and there is no discussion in the manuscript if it is an issues and how it was addressed.

In line 108, the mean should be reported with the standard deviation or the median and the interquartile range.

In lines 134 – 138 of the manuscript, the medians and not the means were reported. Please ensure consistency with line 108.

In line 139 table 1. The table needs formatting to ensure clarity. Are the values for neonatal transport part of the Admission group ( the values under the category admission did add up to the total). It is not obvious from the table, which values neonatal transport, and Treatment falls under or how they each sum to the total?

In line 181 “… decreased by -68% and -45%,…” should be corrected to “… decreased by 68% and 45%,…”

Inline 181 do the authors mean “… slowly improving….”? The sentence should be revised for easy understanding.

6. PLOS authors have the option to publish the peer review history of their article (what does this mean?). If published, this will include your full peer review and any attached files.

Reviewer #1: No

Reviewer #2: **Yes: **Robert. B. Penfold

Reviewer #3: No

---

## [Author Response · Author response to Decision Letter 0]

26 May 2021

Response to the comments of reviewer and editor 

Thank you for submitting your manuscript to PLOS ONE. After careful consideration, we feel that it has merit but does not fully meet PLOS ONE’s publication criteria as it currently stands. Therefore, we invite you to submit a revised version of the manuscript that addresses the points raised during the review process.I am returning your manuscript with three reviews. Please pay attention to the following reviewer suggestions and give them due consideration. Please take the following points particularly into account: 

A thorough language revision of the manuscript is needed to ensure clarity and consistency. Please also double-check any numbers and calculations presented

Please indicate the form of consent obtained (written/oral) or the reason that consent was not obtained

Consider including measures such as neonatal mortality, gestational age, birth weight and diagnosis of neonatal encephalopathy in the analysis; if not please state why not

Methods should be reported in accordance with the Strengthening the Reporting of Observational Studies in Epidemiology (STROBE) statement

A discussion on the negative impact on infant outcomes is warranted

Please elaborate more on the limitations of the chosen analytical method e.g. seasonal considerations

General Response: We thank the editor for the efforts and thank all the reviewers for their suggestions, which significantly improve the quality of our manuscript. We have followed all the reviewers’ suggestions to revise our manuscript point by point. The manuscript has been polished by an English language editing company. We have also carefully checked the manuscript (AJE). The oral consent was obtained prior to the preprint and initial submission of the manuscript. We have added the measures of gestational age and birth weight in the analysis. We have followed the STROBE guidelines for reporting observational studies. And a discussion on the negative impact on infant outcomes has been provided. Please see our detailed point by point responses below

Response to the comments of Editor

Comment 1：A thorough language revision of the manuscript is needed to ensure clarity and consistency. Please also double-check any numbers and calculations presented

Response: According to the reviewer’s suggestions, the manuscript has been polished by an English language editing company. We have also carefully checked the grammar errors in the revised manuscript (AJE). 

Comment 2：Please indicate the form of consent obtained (written/oral) or the reason that consent was not obtained

Response: The oral consent was obtained prior to the preprint and initial submission of the manuscript. 

Comment 3：Consider including measures such as neonatal mortality, gestational age, birth weight and diagnosis of neonatal encephalopathy in the analysis; if not please state why not

Response: We have added the description in the table1 "gestational age and birth weight" in Results section.No significant differences in gestational age or birth weight were observed between groups. The neonatal mortality rate increased by 0.09% during the COVID‐19 outbreak than the pre‐COVID‐19 period (Page12, Lines 189--191). Although the volume of admissions of newborns that suffering from neonatal encephalopathy was decreased, the results showed an elevated proportion of therapeutic hypothermia during NPIs (Page12, Lines 188--189).

Comment 4：Methods should be reported in accordance with the Strengthening the Reporting of Observational Studies in Epidemiology (STROBE) statement

Response: According to the reviewer’s suggestion, we have followed the STROBE guidelines for reporting observational studies (Page 4-7, Lines 69--145).

Comment 5：discussion on the negative impact on infant outcomes is warranted

Response: According to the reviewer’s suggestion, we have provided a deep discussion on the negative impact of the COVID-19 pandemic on newborn medical behaviors in the revised manuscript (Page14- 15, Lines 232--248). 

Firstly, the results showed an elevated proportion of therapeutic hypothermia during NPIs. Some pregnant women reduced the frequency of antenatal visits due to the fear of contracting COVID-19.Therefore, some potential risk factors may not have been detected in a timely manner, leading to fetal distress and necessitating emergency delivery. Secondly,there was some evidence that the delay of accessing the tertiary care hospital by parents may have resulted in additional laboratory investigations, increases in the days of hospitalization, neonatal intensive care unit admission, and/or death.

Comment 6：Please elaborate more on the limitations of the chosen analytical method e.g. seasonal considerations

Response: To elaborate the limitation of the interrupted time series regression analyses, we have added the contents in the limitation section (Page15- 16, Lines 259--267). 

Interrupted time series regression analyses are regarded as the strongest "quasi-experimental" approach and particularly useful when an RCT is infeasible. This finding has to be interpreted with caution because the study design did not investigate seasonal effects. Use of many treatments varies seasonally because of cyclic variations in the illnesses for which they are prescribed. Accounting for seasonally correlated errors usually requires at least 24 monthly data points. Interrupted time series regression analyses models we discussed assume a linear trend in the outcome within each segment. The assumption of linearity often may hold only over short intervals. Changes may follow non-linear patterns1.

Reference:

1.Wagner AK, Soumerai SB, Zhang F, Ross-Degnan D. Segmented regression analysis of interrupted time series studies in medication use research. J Clin Pharm Ther. 2002; 27:299-309. https://doi.org/10.1046/j.1365-2710.2002.00430.x PMID: 12174032

Replies to Reviewer 1

Comment 1: These represent interesting descriptive data, however would be much richer if measures such as neonatal mortality, gestational age, birth weight and diagnosis of neonatal encephalopathy were included. It would be unusual for these measures, in particular mortality, not to be available from routine hospital data and therefore it seems odd not to have reported these. For instance the authors comment in the discussion that medical complexity and severity increased in those admitted in the second time period – but provide no data with which to show this. Tantalisingly they also comment that rates of therapeutic cooling increase but do not comment on diagnostic rates of neonatal encephalopathy (a sentinel marker of quality of perinatal maternal care). Please clearly state available measures from the HIS and why you have chosen the ones you have chosen.

 Response: We have added the description in the Table1 "gestational age and birth weight,neonatal encephalopathy, neonatal mortality" in Results section.

Comment 2.Authors need to rewrite sections of the methods to describe more clearly the setting. They should not assume understanding or prior knowledge of this region, especially as lockdown and infection rates have been vastly different across the globe.

Response: According to the reviewer’s suggestion, we have rewritten sections of the methods to describe more clearly the setting.

The Department of Neonatology of the Children's Hospital of Chongqing Medical University is a tertiary care hospital and the most extensive local neonate referral center in southwestern China. This facility offers a comprehensive range of services to the districts of Chongqing and neighboring provinces, including Guizhou, Sichuan, Yunnan, and Tibet, and serves approximately 350,000 babies in the associated area. 

Comment 3:I think this paper could be improved enormously with inclusion of broader measures of newborn diagnosis and outcomes.

Response: We have added a newborn diagnosis in Results section(Neonatal encephalopathy ).

Neonatal encephalopathy (NE) is a clinically defined syndrome of disturbed neurologic function in an infant born at or beyond 35 weeks of gestation, manifested by a subnormal level of consciousness or seizures, and often accompanied by difficulty with initiating and maintaining respiration and depression of muscle tone and primitive reflexes1-3. The outcomes following neonatal encephalopathy include death and neurological disabilities such as cerebral palsy, epilepsy and cognitive impairment4, 5.

Reference:

1.Executive summary: Neonatal encephalopathy and neurologic outcome, second edition. Report of the American College of Obstetricians and Gynecologists' Task Force on Neonatal Encephalopathy. Obstetrics and gynecology 2014; 123(4): 896-901.

2. Dixon B, Reis C, Ho W, Tang J, Zhang J. Neuroprotective Strategies after Neonatal Hypoxic Ischemic Encephalopathy. International journal of molecular sciences 2015; 16(9): 22368-401.

3. Edwards A, Brocklehurst P, Gunn A, et al. Neurological outcomes at 18 months of age after moderate hypothermia for perinatal hypoxic ischaemic encephalopathy: synthesis and meta-analysis of trial data. BMJ (Clinical research ed) 2010; 340: c363.

4. Hintz S, Barnes P, Bulas D, et al. Neuroimaging and neurodevelopmental outcome in extremely preterm infants. Pediatrics 2015; 135(1): e32-42.

5. Doyle L, Roberts G, Anderson P. Outcomes at age 2 years of infants < 28 weeks' gestational age born in Victoria in 2005. The Journal of pediatrics 2010; 156(1): 49-53.e1.

Comment 4:Methods need to set the scene much more clearly – for example, what is the population size covered by this unit (which from the description of newborn interventions provided would not be described as low resource). What is the usual care fatality rate for this unit? How premature are the babies who are cared for? What is the newborn birth rate within the hospital? It is not clear at all what admissions from outpatients refers to – please describe this more clearly. Anecdotally from my own work in Hong Kong a number of babies with prolonged jaundice and poor weight gain were admitted via outpatients to the NICU – not typical of UK settings. Is this similar in your hospital?

Response: We agree and we have tried to mitigate this problem by fully rewrite sections of the methods.

This facility offers a comprehensive range of services to the districts of Chongqing and neighboring provinces, including Guizhou, Sichuan, Yunnan, and Tibet, and serves approximately 350,000 babies in the associated area. The Department of Neonatology has 310 beds and provides services that account for nearly 10,000 hospital admissions each year over the last 5 years. 

Neonatal mortality rate 1.03% in Pre‐COVID period while neonatal mortality rate 1.12% in COVID impacted period.

In Pre-COVID period study, 1.65% of the newborn (n=201)had a gestational age less than 28 weeks,5.93% (n=705)had a gestational age of 28+0–31+6 weeks and 23% (n=2775)with a gestational age of 32+0–36+6 weeks. 

In COVID impacted study,1.63% of the newborn (n=74)had a gestational age less than 28 weeks,5.48% (n=249) had a gestational age of 28+0–31+6 weeks and 23.30% (n=1056)with a gestational age of 32+0–36+6 weeks. 

Babies can be admitted to this neonatal unit from 2 sources: outborn (transfer from other medical facilities and emergence) and outpatients (outborn).

The most common causes of outpatients included Neonatal pneumonia, pathological early jaundice and prolonged jaundice. Gestational age was near-term and full-term. 

We recruited critically ill neonates from other medical facilities and emergence. Such as neonatal respiratory distress syndrome(NRDS), meconium aspiration syndrome(MAS), Hypoxic-ischemic encephalopathy(HIE), very low birth weight (<1500 g)/ low birth weight (1500–2499 g), preterm and so on.

Comment 5:should be reported in accordance with the Strengthening the Reporting of Observational Studies in Epidemiology (STROBE) statement.

Response:We have followed the STROBE guidelines for reporting observational studies(Page 4-7, Lines 69--145)..

Comment 6:.Also authors need to set the scene in the methods describing what policies were put in place by the local hospital systems due to COVID – for example, were hospitals told to reduce transfers? What were the pathways to admit from a referring hospital? Do you have isolation rooms etc? what lockdown/ NPIs were put in place and how did these change over time after the 24th? What are the staffing levels in the unit? How many ventilated beds/cots do you have? Did that change during the lockdown period?

Response: We have described the hospital systems policies in the methods.

The Chongqing government initiated the first-level response to major public health emergencies and the implementation of the NPIs on January 24, 2020. After the start of the NPIs, the hospital established a special isolation ward that was equipped with 8-16 beds, 5-8 trained nurses, and 2 pediatricians. The ward was equipped with a ventilator including invasive and noninvasive (CPAP and NIPPV) ventilation. During the pandemic, there was no requirement to reduce neonatal .

Comment 7:Results comment around proportion of emergency and inter-hospital transfers increasing (lines 132) is confusing. Would be easier to understand if the methods included a brief description of where babies can be admitted from, eg. "babies can be admitted to this neonatal unit from 3 sources: labour ward/maternity care (inborn), outborn (transfers in from other hospitals) and outpatients (outborn)."

Response: Thank the reviewer’s suggestion. Babies can be admitted to this neonatal unit from 2 sources: outborn (transfer from other medical facilities and emergence) and outpatients (outborn). We have included this description in the methods of the revised manuscript.

Comment 8:Univariate analysis appear to have been unnecessarily repeated – eg table 1 – chi2 should be conducted for one group of categorical outcomes rather than for each individual one. Please only present a single p value therefore for the group.

Response: As the reviewer suggested, we have amended the table in the revised maunuscript (Page 8, table 1).

Comment 9:Would have been useful to present % increase or decrease by week of admission rates for example.

Response: As suggested by the reviewer, we have presented the percent of increase or decrease by week of asmission rates in the revised manuscript.

The admissions per week of neonates dropped by 53%. The number of admissions continued to decline until it reached a nadir of 108 babies/week (Maximum number of hospital admissions 268 babies/week, dropped by 60 %). 

Comment 10:Results – major concern is exclusion of case fatality rate data and diagnosis of neonatal encephalopathy.

More detail can be given to jaundice- were these increases in pathological early jaundice, physiological jaundice or prolonged jaundice?

Response: Fatality rate data and neonatal encephalopathy has been added to the Results section.

Jaundice refers to Pathological jaundice-related diseases(Such as : pathological jaundice;Neonatal hyperbilirubinemia; Hemolytic Disease of the Newborn;Glucose-6-phosphate dehydrogenase (G6PD) deficiency;bilirubin encephalopathy and so on).

Comment 11:Discussion needs editing to explore data in more detail. For example what could the impact of this been on the wider health economy? did more babies die? What is the projected change in admissions like over time? When do you expect to reach previous levels? Or will you? Has the healthcare delivery in other settings/ home improved? Were some of the previous admissions inappropriate/unnecessary? More thought needs to be given to potential generalisability eg lines 201-202 – this is simply not true of the typical newborn population and likely related to the fact you are a tertiary surgical centre.

Response: The results showed a slow recovery of increase in the admissions per week of the neonatal department. Nevertheless, more time is required to return to the pre-COVID period levels of admissions. After 1 year (January 24, 2021), the admissions per week of the neonatal has not recovered to pre-COVID period levels.

The neonatal mortality rate increased by 0.09% during the COVID‐19 outbreak compared with the pre‐COVID‐19 period (Table 1).

I agree fully with the reviewers comments. The possible reason may be overly liberal criteria for admitting patients in pre-COVID, some of which may not have been necessary. However, this COVID‐19 outbreak may indirectly reflect that the pandemic elevated the threshold for inpatient hospitalization.

This study was derived from a single tertiary care center with a retrospective design and a relatively small sample size. Therefore, it does not have strong universal validity, especially for primary health care.

We agree with the reviewer that the COVID-19 pandemic has had an unprecedented effect on health care and the economy, and this analysis will be carried out in the future elsewhere.

Comment 12:Discussion needs considerable rewriting to revisit some of the statements that are not evidenced through the data presented here.

Line 219 – I would argue that this is the most interesting data from this paper and least explored. As discussed above – what were the rates of neonatal encephalopathy diagnosis/ of deaths? Can you comment on reduced ventilated babies – is this because there were fewer premature babies (something reported in other countries) – can you test that within your data by looking at gestational age on admission?

Lines 223-224 – what is your evidence to back this statement up?> how have medical resources been more efficiently allocated? Might you want to continue this efficient allocation in the future (assuming you can evidence this statement). How has this allocation of resources impacted upon neonatal mortality and morbidity and on parental experience of healthcare? And on maternal outcomes? Were you able to measure and report on stillbirth rates for example?

Lines 224-225 – you don't present any data to back this statement up in this paper. I have no doubt that has happened but it appears outkeeping with the data.

Lines 227-230 – you do not present any evidence to back this statement up at all.

Lines 231-232 – this is likely true, but how do your data help with this? Admission rates alone are not sufficient measure (and in fact one could argue an extremely poor measure) of efficacy of the healthcare system.

Lines 232-233 – this is also undoubtedly true – but how does this relate to your data?

Lines 238-239 – arguably this is a moderately well resourced unit – more detail needs ot be provided to allow the reader to compare resource of this unit to other settings.

Lines 241-243 – not really, you don't show change in newborn behaviour.

Need to explore the what next? What additional data would be useful here with which to explore some of these hypotheses? For example from the wider health economy/ qualitative data from parents/ healthcare workers?

Response: Thank you for your advice. After a detailed discussion of the reviewer's suggestion, we are in agreement with your suggestion and have removed it.

Comment 13:Conclusion – poorly written

Response: According to your suggestion,the Conclusion section was rewritten(Page 16, Lines 270-277).

Comment 14:General comments

Newborns arguably are unable to behave in any particular way – rather the healthcare systems and their parents/carers/ maternity services vary. Please rephrase "newborn healthcare behaviour" (used throughout the paper).

Ensure term pandemic rather than epidemic is used throughout

Response: It should be “newborn healthcare behaviour and pandemic” and appropriate correction has been made in the revised manuscript.

Minor comments

Comment 1:Long and short title need to state location

Response: COVID-19 pandemic on newborn medical behaviors

Comment 2:.Abstract

• background –needs to state where this study has been conducted

• methods – needs to include the time periods and which lockdown interventions and when

• methods - Define what the pre-COVID and COVID-impacted periods are. Seems pre-COVID period is very short

• conclusion – insufficient – needs to comment on fact that admissions reduced dramatically, these data only capture a small part of the picture – what happened to the non-admitted babies – did overall infant mortality rate increase for example?

Response:We thank the reviewer for the very helpful suggestions and advice. We have rewritten the part of the Abstract the reviewer found problematic.

Comment 3:.Introduction

More information on background literature needs to be included here. For example a modelling study of this threat predicted between 250000-1.1 million extra neonatal deaths as a result of decreased service provision and access in LMICs. https://www.medrxiv.org/content/10.1101/2021.01.06.21249322v1

Response: While newborns are at low risk for severe disease and death from COVID-19, any impacts on their health outcomes will likely be attributable to the indirect effects of the pandemic on health systems1,2.

Reference:

1.Liguoro I, Pilotto C, Bonanni M, Ferrari ME, Pusiol A, Nocerino A, et al. SARS-COV-2 infection in children and newborns: a systematic review. Eur J Pediatr. 2020; 179:1029-46. https://doi.org/10.1007/s00431-020-03684-7 PMID: 32424745

2. Castagnoli R, Votto M, Licari A, Brambilla I, Bruno R, Perlini S, et al. Severe Acute Respiratory Syndrome Coronavirus 2 (SARS-CoV-2) Infection in Children and Adolescents: A Systematic Review. JAMA Pediatr. 2020; 174:882-9. https://doi.org/10.1001/jamapediatrics.2020.1467 PMID: 32320004

Comment 4:Introduction

Line 51 – "in" not "since"

Line 55 – "caused" not "causes"

Line 60 – consider abbreviation for NPI – non pharmaceutical interventions and then use this rater than lockdown throughout

Line 63 "describing .. changes in newborn care" rather than "on the.."

Line 66 – see comment re phrase newborn medical behaviour being in appropriate for this age group

Methods

Line 71 – do you mean 10,000 babies per year over the last 5 years? Or 10,000 over the last 5 years? Unclear, please edit for sense.

Line 74 – no need for "besdies"

Line 91 – see comment rephrase newborn medical behaviour being in appropriate for this age group

Line 112 – I would not reloy on differentiating btween p values 0.05 or 0.01.

Methods- please describe how and when the diagnoses were given to the babies and by whom.

Response: Correction has been made in the revised manuscript. 

Comment 5:Results

Line 123 – replace visits with admissions

Lines 121-127 – please use phrase admissions per week (if this is what you are referring to) and edit this section for sense. For example – phrase "then it was reduced to a nadir of 108 people…" is not clear – do you mean it reduced to an admission rate of 108 babies/ week or that there were only 108 babies resident on the unit for that particular week?

figure 1 – please include dates on x axis, and also edit y axis to more accurately represent this measure (which I think is admissions per week)

figure 1 – annotate the figure to include what the dotted line indicates rather than just including this in the footnotes

line 132 – pandemic, not epidemic

line 132 – do not jump to causality here.

Lines 155-156 do not make sense

Lines 156-157 –when did a relatively high propotion of babies present with jaundice related conditions?

Figure 2 – please clarify and annotate as per figure 1

Response: Correction has been made in the revised manuscript. 

Comment 6:Discussion

Lines 177-178 edit for sense. Also highlight that these other cohorts do not include newborn data

Line 182 – pandemic not epidemic

Line 182 – edit to read admissions rather than entries

Line 195 – please include this detail in the methods

Line 206 – be more specific here between physiological/ pathological jaundice

Line 217 – how might you have measured degree of severity?

Line 218 – pandemic not epidemic

Response:This comment has been accepted. The correction has been made in the revised manuscript.

Replies to Reviewer 2

Comment 1:While the study is statistically impeccable and the writing is very clear, more discussion of the negative impact on infant outcomes seems warranted. Similarly, if there has not been a commensurate increase in negative outcomes does that suggest many NICU admissions are unnecessary? The decrease in volume by itself is not that interesting. One approach might be a parallel ITS of negative outcomes with the same interruptions specified.

Response: We have analysis the negative impact of the COVID-19 pandemic on newborn medical behaviors in the discussion section (Page 14-15, Lines 232--248). 

Replies to Reviewer 3

Comment 1:In the abstract the authors stated that "Overall, 13,540 infants were admitted to the NICU during the pre-COVID period (n= 12082) and COVID-impacted period (n=4558)." The figures provided in line 34 – 35 do not add up and the statement is not clear on the disparity between the 13,540 and the figures provided in parentheses. It is also not clear what NICU means at this stage.

Response:We are very sorry for our incorrect writing that 'Overall, 13,540 infants were admitted to the NICU', according to your suggestion, we have re-written this part in the abstract section.

Overall, 16,640 infants were admitted to the neonatology department during the pre-COVID period (n = 12,082) and the COVID-impacted period (n =4,558). 

Comment 2:In line 64 it seems that the impact of lockdown measures ( such as curfew, opening and closing hours of business, who and where one could travel to ) on neonatal admissions and disease spectrum are to be studied. However, that seems not to be the case. The analysis and results presented seem to have studied if there is changes in neonatal admission and diseases spectrum pre-COIVD and during the COVID period. Or are the authors referring to the period of lockdown as the "lockdown measures"? This should be clarified in the manuscript.

Response: The article was to detect changes in the levels and trends of hospital admissions and disease spectra before and after the period of lockdown measures and assess the impact of the control measures.

Comment 3:In line 77 it was stated the "clinical and demographics information were collected…" while in line 78 though attempt was made to list the clinical information but the lists provided in line 79 does not differentiate between clinical and demographic information.

Response:Correction has been made in the revised manuscript. 

The variables selected to describe the demographic and clinical information of the cases were the volume of admissions and neonate transport, sex, age (days), admission, length of hospital stay (days), treatment, neonatal mortality, gestational age (weeks + days), birth weight (gram) and disease spectra.

Comment 4:The authors should consider formatting the date on line 84 and 85 to commonly used time format for example (yyyy/m/d – yyyy/m/d).

Response: The format of the date has been changed in the revised manuscript.

Comment 5:In the manuscript in line 87 to 91 the outcomes and dependent variables were not clearly stated. Clarity is needed for the readers.

Response: The outcome variable was the weekly number of cause-specific hospital admissions. The dependent variable was time, intervention and Post-slope.

Comment 6:The use of interrupted time-series and the choice of break point seem appropriate. However, the authors should consider a change point analysis to detect a change in the data generating process. This might as well provide the basis for choosing or confirming the choice of the break point used in this analysis. In addition, time series analyses may be confounded by seasonal effects and there is no discussion in the manuscript if it is an issues and how it was addressed.

Response: To elaborate the limitation of the interrupted time series regression analyses, we added the contents in the limitation section (Page15- 16, Lines 259--267). 

We agree with the reviewer that change point analysis to detect a change in the data generating process, and this analysis will be carried out in the future elsewhere.

Comment 7:In line 108, the mean should be reported with the standard deviation or the median and the interquartile range.

In lines 134 – 138 of the manuscript, the medians and not the means were reported. Please ensure consistency with line 108.

Response:We apologize for this mistake,the correction has been made in the revised manuscript(Page 7, line 138).

Comment 8:In line 139 table 1. The table needs formatting to ensure clarity. Are the values for neonatal transport part of the Admission group ( the values under the category admission did add up to the total). It is not obvious from the table, which values neonatal transport, and Treatment falls under or how they each sum to the total?

Response: As reviewer suggested,the table have been amended to make it as simple as possible to understand(Page 8, table 1).

Comment 9:In line 181 "… decreased by -68% and -45%,…" should be corrected to "… decreased by 68% and 45%,…"

Response: Appropriate correction has been made in the revised manuscript(Page 13, line 201

 ).

Comment 10:Inline 181 do the authors mean "… slowly improving…."? The sentence should be revised for easy understanding.

Response: With the abatement of the pandemic, the neonatal admission volume slowly recovered. 

Special thanks to you for your good comments. 

We tried our best to improve the manuscript and made some changed in the manuscript. These changes are marked in red in revised paper.

We appreciate for editors' and reviewers' warm work earnestly, and hope that the correction will meet with approval.

Once again, thank you very much for your comments and suggestions.

Best regards

Sincerely

Corresponding author: Zi-Yu Hua

E-mail: h_ziyu@126.com

---

## [Decision Letter · Decision Letter 1]

9 Aug 2021

PONE-D-21-02629R1

The impact of the COVID-19 pandemic on neonatal  admissions: an interrupted time-series study

PLOS ONE

Dear Dr. Hua,

Thank you for submitting your manuscript to PLOS ONE. After careful consideration, we feel that it has merit but does not fully meet PLOS ONE’s publication criteria as it currently stands. Therefore, we invite you to submit a revised version of the manuscript that addresses the points raised during the review process.

Please answer, all the comments of the reviewers. As they highlight, there are several points that need to be addressed mainly in the data description and results interpretation.  The results (for example, table 1) would be easier to be interpreted if presented as rate per week. Otherwise, the values by disease groups, treatment, mortality and others produce confusion. Besides, to observe the variation in neonatal mortality the change in case fatality rate/ week over time should be added.

We look forward to receiving your revised manuscript.

Kind regards,

Juan F. Orueta, MD, PhD

Academic Editor

PLOS ONE

Reviewers' comments:

Reviewer's Responses to Questions

**Comments to the Author**

1. If the authors have adequately addressed your comments raised in a previous round of review and you feel that this manuscript is now acceptable for publication, you may indicate that here to bypass the “Comments to the Author” section, enter your conflict of interest statement in the “Confidential to Editor” section, and submit your "Accept" recommendation.

Reviewer #1: (No Response)

Reviewer #2: All comments have been addressed

Reviewer #3: All comments have been addressed

2. Is the manuscript technically sound, and do the data support the conclusions?

Reviewer #1: No

Reviewer #2: (No Response)

Reviewer #3: (No Response)

3. Has the statistical analysis been performed appropriately and rigorously? 

Reviewer #1: No

Reviewer #2: (No Response)

Reviewer #3: Yes

4. Have the authors made all data underlying the findings in their manuscript fully available?

Reviewer #1: No

Reviewer #2: (No Response)

Reviewer #3: (No Response)

5. Is the manuscript presented in an intelligible fashion and written in standard English?

Reviewer #1: No

Reviewer #2: (No Response)

Reviewer #3: Yes

6. Review Comments to the Author

Reviewer #1: Whilst I thank the authors for their efforts to address my comments, I do not believe they have sufficiently addressed my concerns/comments nor do they describe the data sufficiently well – either in their analysis or in their interpretation and discussion. For example, they should look to describe change in case fatality rate/ week over time to show how that varies before and after introduction of NPIs. Of note, the figures show drops in admissions from all causes, with recovery only apparent from pathological jaundice.

Other comments:

Please state where this study is conducted in both the long and short title – not addressed.

• Long and short title need to state location eg The impact of the COVID-19 pandemic on neonatal admissions to a tertiary surgical neonatal unit, Southwest-central China: an interrupted time-series study

Please rephrase “newborn medical behaviours” throughout (not addressed)

• “newborn medical behaviors” – please use a better phrase to describe the delivery of newborn care following the pandemic and/or parental care seeking behaviours. As noted previously, newborns do not seek care – their parents/ the healthcare system behaviour determines their care. Babies do not seek consultations, their parents/ caregivers seek medical review.

Abstract,

• Background – state “China” in location as many will not know where Chongqing Medical University is.

• Results section appear to be contradictory and confusing – please edit for sense. This section does not make sense “There was no decline in the volume of admissions of newborns suffering from respiratory system diseases, infectious diseases, and gastrointestinal diseases immediately after the COVID-impacted period (p =0.08, p =0.11, p =0.06, respectively). There was an immediate decline in the volume of patients complaining about pathological jaundice-related disease conditions” – however table 1 and figures show casing the data suggests falls in all of these admissions.

Setting

• It appears this neonatal unit does not accept direct referrals from a maternity unit/ labour ward? Is this correct? Please clarify and add this detail to the methods. What does the emergency category pertain to?

• What proportion of the admissions are usually from out of province?

Results

• “The patients seeking consultations in the COVID-impacted period were younger than those seeking consultations in the pre-COVID period 159 (median age 5 days vs. 6 days, p<0.05).” – this is likely because those admitted via outpatients were reduced – if you compare against age in those admitted from interhospital transfers only is there a difference in age?

• Where is the p value for the mortality rate in table 1? Please present rate not percentage for a mortality rate (typically presented as case fatality rate of X babies per 1000 admitted babies) – here table 1 suggests a change from 10.3 per 1000 to 11.2 per 1000 – a 10% increase in mortality (not a 0.09% increase). Please clarify this on page 26 line 189-190.

• Data should be presented as weekly rates – for example in the text include average weekly admission rates of 220/week pre COVID and 147/week post COVID; average weekly admissions for respiratory disorders pre and post decreased from 76/wk to 48/wk; admissions for neonatal encephalopathy from 4/wk to 2.5/wk – of those admitted with NE – 23% required therapeutic hypothermia compared to 56% post covid – suggesting those babes were sicker on admission.

• Weekly rates of pathological jaundice admissions dropped from 59/week to 46/week – but that drop was not as substantial as other disorders, therefore relatively speaking the proportion of admissions attributable to jaundice increased, but the absolute number reduced.

Discussion

• Needs discussion on change in mortality rate

• This statement “This is partly because gastrointestinal disorders are relatively more common in newborns due to inborn anomalies (e.g., megacolon, esophageal atresia, pyloric hypertrophic obstruction)[19]” is not correct. It may be that in your surgical unit these disorders are relatively more common, however in the typical neonatal population these are most certainly NOT more common than respiratory disorders/ infections.

• This statement is not true “Almost all newborns develop pathological hyperbilirubinemia[20, 21]” furthermore references 20 and 21 relate to clinical assessment of jaundice rather than rates of pathological jaundice.

• Need to consider impact on entire health economy – eg did more mother’s deliver at home? What was the change in stillbirth rate? The change in mortality rates in the secondary care units? The discussion presents a far too simplistic discussion around potential explanations for the changes in rates and diagnosis documented.

• Discussion of change in therapeutic hypothermia focuses entirely on antenatal access – by mothers – not at all on availability of these services due to health service changes nor to potential discussion of quality of care available during the perinatal period – this needs to be addressed as it does not adequately address potential factors involved in resultant NE, but instead focuses on maternal factors in their behaviour of accessing health services.

• Lines 242-245 – the data don’t reflect this. Too much weight is placed on parent behaviours.

• Limitations are not sufficiently well described – of note this is a surgical tertiary referral unit that doesn’t appear (from the description) to admit directly from maternity services – extremely unusual for most neonatal units across the globe.

Reviewer #2: All of my concerns have been addressed.

...........................................................

Reviewer #3: Dear authors,

I am happy that my points were taken into considerations and I appreciate the authors for the overall improvement of the manuscript.

However, I have minor suggestions that might be useful for both the author and research community.

In line 144 – 145, the author cited the R software, from the citation (I assume that the package used were R based package), if the packages are not “R base” the author should fully cite the packages if not shipped as part of R base functions. The citation of the packages acknowledges the effort and time spent by the people that created these tools and help in reproducibility of the results.

All citation should also appear in the reference list.

For example, using and citing RStudio and lme4 package assuming both are the 18th and 19th cited materials:

All computations were conducted in RStudios 1.0.44 (18) … with lme4 v.xxx (19)…. While in the reference list the citation for lme4 should appear as:

Douglas Bates, Martin Maechler, Ben Bolker, Steve Walker (2015). Fitting Linear Mixed-Effects Models Using lme4. Journal of Statistical Software, 67(1), 1-48. doi:10.18637/jss.v067.i01.

R Core TEAM(2014). R. A language and environment for statistical computing. R Foundation for Statistical Computing, Vienna, Austria. URL http://www.R-project.org/

Table 1, in line 165 need a little adjustment. The section for weeks + day, require addition information. For instance 201(1.65), does it represent mean(sd) or median(IQR). Please let it be consistent like the section on Birth weight(gram), median(IQR), and the section on Age which has median (IQR) etc.

7. PLOS authors have the option to publish the peer review history of their article (what does this mean?). If published, this will include your full peer review and any attached files.

Reviewer #1: No

Reviewer #2: **Yes: **Robert B. Penfold

Reviewer #3: No

---

## [Author Response · Author response to Decision Letter 1]

3 Oct 2021

Response to the comments of Editor

Comment 1：Please answer, all the comments of the reviewers. As they highlight, there are several points that need to be addressed mainly in the data description and results interpretation.  The results (for example, table 1) would be easier to be interpreted if presented as rate per week. Otherwise, the values by disease groups, treatment, mortality and others produce confusion. Besides, to observe the variation in neonatal mortality the change in case fatality rate/ week over time should be added.

Response: We agree and we have tried to mitigate this problem by fully rewrite sections of the results( table 1) .We have added the description in the table1 "fatality rate/ week " in Results section(Page 8, Lines 169).

Replies to Reviewer 1

Comment 1:For example, they should look to describe change in case fatality rate/ week over time to show how that varies before and after introduction of NPIs. Of note, the figures show drops in admissions from all causes, with recovery only apparent from pathological jaundice.

Response:According to the reviewer’s suggestion, We have added the description in the table1 "fatality rate/ week " in Results section(Page 8, Lines 169).

There was an evident decrease in the volume of admissions/week for all disease spectra after the intervention. Additionally, there was a statistically significant decrease in the number of patients with pathological jaundice-related conditions(p<0.05)(Table 2 and Figure 2).

However, their recovery trend was faster than that other disorders (p<0.05), therefore relatively speaking the proportion of admissions attributable to jaundice increased.

Other comments:

Comment 2.Please state where this study is conducted in both the – not addressed.

Response: Correction has been made in the revised manuscript. 

long title: Impact of the COVID-19 Pandemic on neonatal admissions in a tertiary Children's Hospital in Southwest China: an interrupted time-series study.

short title:COVID-19 pandemic affecting neonatal admissions in southwest China

Comment 3:Please rephrase “newborn medical behaviours” throughout (not addressed)

Response: It should be“health-seeking behavior of parents/caregivers.”and appropriate correction has been made in the revised manuscript.

Abstract

Comment 4:Background – state “China” in location as many will not know where Chongqing Medical University is.

Response: We thank the reviewer for the very helpful suggestions and advice. We have rewritten the part of the Abstract the reviewer found problematic.

This study was conducted to investigate the impact of lockdown measures in a tertiary Children's Hospital in southwest China, which might be used to predict long-term effects related to health-seeking behavior of parents/caregivers(Page 1, Lines 20-22).

Comment 5:Results section appear to be contradictory and confusing – please edit for sense. This section does not make sense “There was no decline in the volume of admissions of newborns suffering from respiratory system diseases, infectious diseases, and gastrointestinal diseases immediately after the COVID-impacted period (p =0.08, p =0.11, p =0.06, respectively). There was an immediate decline in the volume of patients complaining about pathological jaundice-related disease conditions” – however table 1 and figures show casing the data suggests falls in all of these admissions.

Response: We thank the reviewer for the very helpful suggestions and advice. We have rewritten the part of the Abstract the reviewer found problematic(Page 2, Lines 35-43).

The average weekly admission rates of 220/week pre COVID and 147/week post COVID. There was an evident decrease in the volume of admissions for all disease spectra after the intervention, whereas the decrease of patients complaining about pathological jaundice-related conditions was statistically significant (p<0.05). In the COVID-impacted period, the percentage of patients who suffered from respiratory system diseases, neonatal encephalopathy, and infectious diseases decreased, while the percentage of pathological jaundice-related conditions and gastrointestinal system diseases increased. The neonatal mortality rate increased by 9‰ during the COVID-19 outbreak compared with the pre-COVID period.

Setting

Comment 6:It appears this neonatal unit does not accept direct referrals from a maternity unit/ labour ward? Is this correct? Please clarify and add this detail to the methods. What does the emergency category pertain to?

Response: According to the reviewer’s suggestion, we have rewritten sections of the methods to describe more clearly the setting.(Page 4, Lines 90-94).

Dedicated transport teams are staffed separately from NICU personnel specifically for the purpose of transport of patients to and from the hospital. These personnel do not have patient assignments. Our neonatal unit receives direct referrals from the labour ward or Level I or II neonatal care services using the transport system. 

"Emergency" means that critically ill patients were transferred and admitted into our neonatal unit directly via outpatients.

Comment 7:What proportion of the admissions are usually from out of province?

Response:According to the reviewer’s suggestion, We have added the description in the table1 "Proportion of the admissions are usually from out of province " in Results section(Page 8, Lines 169).

 Pre‐COVID COVID impacted p value

Proportion of the admissions are usually from out of province，n (%) 

2225(18.4) 

702(15.4) 

p<0.05

Results

Comment 8:“The patients seeking consultations in the COVID-impacted period were younger than those seeking consultations in the pre-COVID period 159 (median age 5 days vs. 6 days, p<0.05).” – this is likely because those admitted via outpatients were reduced – if you compare against age in those admitted from interhospital transfers only is there a difference in age?

Response: I agree with these concerns, and feel that - age is associated with a higher percentage of in outpatients(outborn).

Comment 9: Where is the p value for the mortality rate in table 1? Please present rate not percentage for a mortality rate (typically presented as case fatality rate of X babies per 1000 admitted babies) – here table 1 suggests a change from 10.3 per 1000 to 11.2 per 1000 – a 10% increase in mortality (not a 0.09% increase). Please clarify this on page 26 line 189-190.

Response: 

Average rate of weekly Fatality Pre‐COVID COVID impacted p value

n (‰) 2.3（10.3） 1.6（11.2） p<0.05

NMRs increased by 9‰ during the COVID‐19 outbreak compared with the pre-COVID. However, no statistical differences were found between NMRs (p=0.66)(Page 13, Lines 193-194).

Comment 10:Data should be presented as weekly rates – for example in the text include average weekly admission rates of 220/week pre COVID and 147/week post COVID; average weekly admissions for respiratory disorders pre and post decreased from 76/wk to 48/wk; admissions for neonatal encephalopathy from 4/wk to 2.5/wk – of those admitted with NE – 23% required therapeutic hypothermia compared to 56% post covid – suggesting those babes were sicker on admission.

Response: We agree and we have tried to mitigate this problem by fully rewrite sections of the results( table 1) .

Comment 11:Weekly rates of pathological jaundice admissions dropped from 59/week to 46/week – but that drop was not as substantial as other disorders, therefore relatively speaking the proportion of admissions attributable to jaundice increased, but the absolute number reduced

Response: We agree and we have tried to mitigate this problem by fully rewrite sections of the results(Page 12, Lines 184-188) .

There was an evident decrease in the volume of admissions/week for all disease spectra after the intervention. Additionally, there was a statistically significant decrease in the number of patients with pathological jaundice-related conditions(p<0.05)(Table 2 and Figure 2).

However, their recovery trend was faster than that other disorders (p<0.05), therefore relatively speaking the proportion of admissions attributable to jaundice increased.

Discussion

Comment 12: Needs discussion on change in mortality rate

Response:According to the reviewer’s suggestion, we have provided a deep discussion on change in mortality rate in the revised manuscript (Page15- 16, Lines 246--267). 

The data suggested that neonatal death might be increased in COVID-impacted period. Preterm birth is the leading cause of neonatal mortality and morbidity in both high- and low-income countries.No significant differences were observed between groups in gestational age. Therefore, this increase in mortality rate appears to be driven by other factors and is not likely to be explained by preterm birth. It is more likely that changes in illness severity, health-seeking behavior of parents/caregivers, and inadequate medical resources factors. 

The admissions for neonatal encephalopathy from 4/wk to 2.5/wk–of those admitted with NE–23% required therapeutic hypothermia compared to 56% post covid–suggesting those babes were sicker on admission. The neonatal period is classified as early (the first 7 days) and late (the remaining 21 days) neonatal period, and majority neonatal mortality occur in the early neonatal period. The proportion of children aged ≤7 days old was higher in the COVID-impacted period(54.7% pre-COVID and 58.8% post-COVID).In China, the implementation of NPIs, also resulting in severe restrictions affecting the health-seeking behavior of parents/caregivers. There was concern that parents/ caregivers stay at home to care for her baby with progressive disease rather than come to the hospital where the risk of contracting COVID-19 was perceived to be higher. The Department of Neonatology has insufficient critical care staff during COVID-impacted period. It is possible that reduced capacity of healthcare services to provide timely diagnoses and treatment. Reassuringly, these mortality analyses found no statistical differences (p=0.66). These results should be interpreted with caution, and additional studies are needed to clarify whether our observations represent a significantly higher mortality rate adjusted for disease severity or whether they merely reflect increased primary care management of less severe conditions.

Comment 13:This statement “This is partly because gastrointestinal disorders are relatively more common in newborns due to inborn anomalies (e.g., megacolon, esophageal atresia, pyloric hypertrophic obstruction)[19]” is not correct. It may be that in your surgical unit these disorders are relatively more common, however in the typical neonatal population these are most certainly NOT more common than respiratory disorders/ infections.

Response: We agree and we have tried to mitigate this problem by fully rewrite sections of the discussion (Page14, Lines 220-225).

However, the proportion of gastrointestinal diseases and surgeries were unaffected by these measures (p=0.21 and p=0.10). This is partly because gastrointestinal disorders, especially anomalies, are relatively more common in our tertiary care hospital (e.g., megacolon, esophageal atresia, pyloric hypertrophic obstruction). Some of these diseases require surgical intervention in the neonatal period. As one of the regional neonate emergency care providers and the most extensive local neonate referral center, only the children's hospital could perform a sequence of operations.

Comment 14:This statement is not true “Almost all newborns develop pathological hyperbilirubinemia[20, 21]” furthermore references 20 and 21 relate to clinical assessment of jaundice rather than rates of pathological jaundice.

Response: Neonatal hyperbilirubinemia is a common condition, affecting more than 60% of full-term and 80% of preterm infants [19, 20, 21]. 

19.Riordan SM, Bittel DC, Le Pichon JB, Gazzin S, Tiribelli C, Watchko JF, et al. A Hypothesis for Using Pathway Genetic Load Analysis for Understanding Complex Outcomes in Bilirubin Encephalopathy. Front Neurosci. 2016; 10:376. https://doi.org/10.3389/fnins.2016.00376 PMID: 27587993

20.Hegyi T, Kleinfeld A, Huber A, Weinberger B, Memon N, Shih W, et al. Unbound bilirubin measurements by a novel probe in preterm infants. J Matern Fetal Neonatal Med. 2019; 32:2721-6. https://doi.org/10.1080/14767058.2018.1448380 PMID: 29504491

21.Gazzin S, Dal Ben M, Montrone M, Jayanti S, Lorenzon A, Bramante A, et al. Curcumin Prevents Cerebellar Hypoplasia and Restores the Behavior in Hyperbilirubinemic Gunn Rat by a Pleiotropic Effect on the Molecular Effectors of Brain Damage. Int J Mol Sci. 2020; 22. https://doi.org/10.3390/ijms22010299 PMID: 33396688

Comment 15:Need to consider impact on entire health economy – eg did more mother’s deliver at home? What was the change in stillbirth rate? The change in mortality rates in the secondary care units? The discussion presents a far too simplistic discussion around potential explanations for the changes in rates and diagnosis documented.

Response: We agree with the reviewer that the COVID-19 Pandemic has drastically impacted on entire health economys, and this analysis will be carried out in the future elsewhere.

Finally, The COVID-19 Pandemic has drastically impacted on entire health economy. For instance, there are potential barriers to accessing obstetric and neonatal health care services, barriers to good healthcare-seeking behavior of parents/ caregivers, transportation barriers, and parents/ caregivers’ financial barriers(housing instability, food insecurity). Further research should be conducted to elaborate on this aspect(Page17, Lines 220-225).

Comment 16: Discussion of change in therapeutic hypothermia focuses entirely on antenatal access – by mothers – not at all on availability of these services due to health service changes nor to potential discussion of quality of care available during the perinatal period – this needs to be addressed as it does not adequately address potential factors involved in resultant NE, but instead focuses on maternal factors in their behaviour of accessing health services.

Response: We agree and we have tried to mitigate this problem by fully rewrite sections of the discussion (Page15, Lines 237-245).

One probable reason for such unacceptable poor quality of maternity care and poor neonatal health outcome is changes in accessing emergency obstetric and /or neonatal cares which include delays in reaching health facilities (delays at the journey) and delays in receiving appropriate care (delays at the health facilities). 

Comment 17:Lines 242-245 – the data don’t reflect this. Too much weight is placed on parent behaviours.

Response: Thank you for your advice. After a detailed discussion of the reviewer's suggestion, we are in agreement with your suggestion and have removed it.

Comment 18:Limitations are not sufficiently well described – of note this is a surgical tertiary referral unit that doesn’t appear (from the description) to admit directly from maternity services – extremely unusual for most neonatal units across the globe.

Response: According to the reviewer’s suggestion, we have rewritten sections of the methods to describe more clearly the setting.(Page 4, Lines 90-94).

Replies to Reviewer 3

Comment 1:However, I have minor suggestions that might be useful for both the author and research community.In line 144 – 145, the author cited the R software, from the citation (I assume that the package used were R based package), if the packages are not “R base” the author should fully cite the packages if not shipped as part of R base functions. The citation of the packages acknowledges the effort and time spent by the people that created these tools and help in reproducibility of the results.

All citation should also appear in the reference list.

For example, using and citing RStudio and lme4 package assuming both are the 18th and 19th cited materials:

All computations were conducted in RStudios 1.0.44 (18) … with lme4 v.xxx (19)…. While in the reference list the citation for lme4 should appear as:

Douglas Bates, Martin Maechler, Ben Bolker, Steve Walker (2015). Fitting Linear Mixed-Effects Models Using lme4. Journal of Statistical Software, 67(1), 1-48. doi:10.18637/jss.v067.i01.

R Core TEAM(2014). R. A language and environment for statistical computing. R Foundation for Statistical Computing, Vienna, Austria. URL http://www.R-project.org/

Response:Correction has been made in the revised manuscript. 

Comment 2:Table 1, in line 165 need a little adjustment. The section for weeks + day, require addition information. For instance 201(1.65), does it represent mean(sd) or median(IQR). Please let it be consistent like the section on Birth weight(gram), median(IQR), and the section on Age which has median (IQR) etc.

Response: We apologize for this mistake,the correction has been made in the revised manuscript(Page 8, Lines 169).

Special thanks to you for your good comments. 

We tried our best to improve the manuscript and made some changed in the manuscript. These changes are marked in red in revised paper.

We appreciate for editors' and reviewers' warm work earnestly, and hope that the correction will meet with approval.

Once again, thank you very much for your comments and suggestions.

Best regards

Sincerely

Corresponding author: Zi-Yu Hua

E-mail: h_ziyu@126.com

---

## [Decision Letter · Decision Letter 2]

22 Oct 2021

PONE-D-21-02629R2Impact of the COVID-19 Pandemic on neonatal admissions in a tertiary Children's Hospital in Southwest China: an interrupted time-series studyPLOS ONE

Dear Dr. Hua,

Thank you for submitting your manuscript to PLOS ONE. After careful consideration, we feel that it has merit but does not fully meet PLOS ONE’s publication criteria as it currently stands. Therefore, we invite you to submit a revised version of the manuscript that addresses the points raised during the review process.

We look forward to receiving your revised manuscript.

Kind regards,

Simone Lolli

Academic Editor

PLOS ONE

Journal Requirements:

Additional Editor Comments (if provided):

The manuscript is ready for publication after taking into consideration the minor changes pointed out by the reviewers.

Reviewers' comments:

Reviewer's Responses to Questions

**Comments to the Author**

1. If the authors have adequately addressed your comments raised in a previous round of review and you feel that this manuscript is now acceptable for publication, you may indicate that here to bypass the “Comments to the Author” section, enter your conflict of interest statement in the “Confidential to Editor” section, and submit your "Accept" recommendation.

Reviewer #1: All comments have been addressed

Reviewer #3: All comments have been addressed

2. Is the manuscript technically sound, and do the data support the conclusions?

Reviewer #1: Yes

Reviewer #3: Yes

3. Has the statistical analysis been performed appropriately and rigorously? 

Reviewer #1: Yes

Reviewer #3: Yes

4. Have the authors made all data underlying the findings in their manuscript fully available?

Reviewer #1: No

Reviewer #3: Yes

5. Is the manuscript presented in an intelligible fashion and written in standard English?

Reviewer #1: Yes

Reviewer #3: Yes

6. Review Comments to the Author

Reviewer #1: Thank-you for addressing the comments. The paper is much clearer and the results and discussion sections in particular are improved. I have only one minor comments:

please ensure the phrase "COVID-impacted period" is used throughout to describe the second time period. In places (eg abstract methods and discussion) it is described as "after covid" or post-covid.

Reviewer #3: I thank the authors for accepting the suggestions and I appreciate how much they greatly improved the manuscript.

I have very few minor comments on editing for the authors to take a second a look at:

Line 124 – stated that is “experiment was designed” or do the author meant that “this study was conducted…”, because I don’t think an experiment was designed in the real sense of it. In line 155 “…. 147/week post-COVID”, do the authors meant COVID-period ?

Lines 250 – 251 “It is more likely that changes in illness severity, health-seeking 251 behavior of parents/caregivers, and inadequate medical resources factors.” , seems incomplete.

Lines 253, 275, 262 “…post COVID…” or do you mean COVID-impacted period ?

7. PLOS authors have the option to publish the peer review history of their article (what does this mean?). If published, this will include your full peer review and any attached files.

Reviewer #1: **Yes: **Dr. Michelle Heys

Reviewer #3: No

---

## [Author Response · Author response to Decision Letter 2]

2 Nov 2021

Response to the comments of Editor

Comment 1：Please review your reference list to ensure that it is complete and correct. If you have cited papers that have been retracted, please include the rationale for doing so in the manuscript text, or remove these references and replace them with relevant current references. Any changes to the reference list should be mentioned in the rebuttal letter that accompanies your revised manuscript. If you need to cite a retracted article, indicate the article’s retracted status in the References list and also include a citation and full reference for the retraction notice.

Response: According to the reviewer’s suggestion, reference list is complete and correct.

Replies to Reviewer 1

Comment 1: please ensure the phrase "COVID-impacted period" is used throughout to describe the second time period. In places (eg abstract methods and discussion) it is described as "after covid" or post-covid.

Response:According to the reviewer’s suggestion, correction has been made in the revised manuscript. 

Replies to Reviewer 3

Comment 1:Lines 250 – 251 “It is more likely that changes in illness severity, health-seeking 251 behavior of parents/caregivers, and inadequate medical resources factors.” , seems incomplete.

Response: We thank the reviewer for the very helpful suggestions and advice. We have rewritten the part of the discussion the reviewer found problematic(Page 15, Lines 251-253 ).

 It is more likely that changes in illness severity, health-seeking behavior of parents/caregivers, and inadequate medical resources factors result in this phenomenon.

Comment 2:Line 124 – stated that is “experiment was designed” or do the author meant that “this study was conducted…”, because I don’t think an experiment was designed in the real sense of it. 

Response: We thank the reviewer for the very helpful suggestions and advice. We have rewritten the part of the methods the reviewer found problematic(Page6 , Lines 124-125).

Second, this study was conducted to evaluate whether the health-seeking behavior of parents/caregivers was influenced by pandemic factors.

Comment 3:Lines 253, 275, 262 “…post COVID…” or do you mean COVID-impacted period ?

line 155 “…. 147/week post-COVID”, do the authors meant COVID-period ?

Response: According to the reviewer’s suggestion, correction has been made in the revised manuscript. 

Special thanks to you for your good comments. 

We tried our best to improve the manuscript and made some changed in the manuscript. These changes are marked in red in revised paper.

We appreciate for editors' and reviewers' warm work earnestly, and hope that the correction will meet with approval.

Once again, thank you very much for your comments and suggestions.

Best regards

Sincerely

Corresponding author: Zi-Yu Hua

E-mail: h_ziyu@126.com

---

## [Decision Letter · Decision Letter 3]

20 Dec 2021

Impact of the COVID-19 Pandemic on neonatal admissions in a tertiary Children's Hospital in Southwest China: an interrupted time-series study

PONE-D-21-02629R3

Dear Dr. Hua,

We’re pleased to inform you that your manuscript has been judged scientifically suitable for publication and will be formally accepted for publication once it meets all outstanding technical requirements.

Kind regards,

Simone Lolli

Academic Editor

PLOS ONE

Additional Editor Comments (optional):

Reviewers' comments:

Reviewer's Responses to Questions

**Comments to the Author**

1. If the authors have adequately addressed your comments raised in a previous round of review and you feel that this manuscript is now acceptable for publication, you may indicate that here to bypass the “Comments to the Author” section, enter your conflict of interest statement in the “Confidential to Editor” section, and submit your "Accept" recommendation.

Reviewer #1: All comments have been addressed

2. Is the manuscript technically sound, and do the data support the conclusions?

Reviewer #1: Yes

3. Has the statistical analysis been performed appropriately and rigorously? 

Reviewer #1: Yes

4. Have the authors made all data underlying the findings in their manuscript fully available?

Reviewer #1: No

5. Is the manuscript presented in an intelligible fashion and written in standard English?

Reviewer #1: Yes

6. Review Comments to the Author

Reviewer #1: Thank-you for addressing all my reviewer comments. I have no additional comments for you to address.

7. PLOS authors have the option to publish the peer review history of their article (what does this mean?). If published, this will include your full peer review and any attached files.

Reviewer #1: **Yes: **Dr Michelle Heys

---

## [Editor Report · Acceptance letter]

5 Jan 2022

PONE-D-21-02629R3 

Impact of the COVID-19 Pandemic on neonatal admissions in a tertiary Children's Hospital in Southwest China: an interrupted time-series study 

Dear Dr. Hua:

I'm pleased to inform you that your manuscript has been deemed suitable for publication in PLOS ONE. Congratulations! Your manuscript is now with our production department. 

Kind regards, 

on behalf of

Dr. Simone Lolli 

Academic Editor

PLOS ONE